# Dissecting the regulatory roles of ORM proteins in the sphingolipid pathway of plants

**Adil Alsiyabi**[1], **Ariadna Gonzalez Solis**[2], **Edgar B. Cahoon**[2], **Rajib Saha**[1,3]*

**1** Department of Chemical and Biomolecular Engineering, University of Nebraska-Lincoln, Lincoln, Nebraska, United States of America, **2** Center for Plant Science Innovation & Department of Biochemistry, University of Nebraska-Lincoln, Lincoln, Nebraska, United States of America, **3** Center for Plant Science Innovation, University of Nebraska-Lincoln, Lincoln, Nebraska, United States of America

* rsaha2@unl.edu

**Data Availability Statement:** All relevant data are within the manuscript and its Supporting Information files.

**Funding:** Funding to support this work was provided by a National Science Foundation grant

## Abstract

Sphingolipids are a vital component of plant cellular endomembranes and carry out multiple functional and regulatory roles. Different sphingolipid species confer rigidity to the membrane structure, facilitate trafficking of secretory proteins, and initiate programmed cell death. Although the regulation of the sphingolipid pathway is yet to be uncovered, increasing evidence has pointed to orosomucoid proteins (ORMs) playing a major regulatory role and potentially interacting with a number of components in the pathway, including both enzymes and sphingolipids. However, experimental exploration of new regulatory interactions is time consuming and often infeasible. In this work, a computational approach was taken to address this challenge. A metabolic network of the sphingolipid pathway in plants was reconstructed. The steady-state rates of reactions in the network were then determined through measurements of growth and cellular composition of the different sphingolipids in Arabidopsis seedlings. The Ensemble modeling framework was modified to accurately account for activation mechanisms and subsequently used to generate sets of kinetic parameters that converge to the measured steady-state fluxes in a thermodynamically consistent manner. In addition, the framework was appended with an additional module to automate screening the parameters and to output models consistent with previously reported network responses to different perturbations. By analyzing the network's response in the presence of different combinations of regulatory mechanisms, the model captured the experimentally observed repressive effect of ORMs on serine palmitoyltransferase (SPT). Furthermore, predictions point to a second regulatory role of ORM proteins, namely as an activator of class II (or LOH1 and LOH3) ceramide synthases. This activating role was found to be modulated by the concentration of free ceramides, where an accumulation of these sphingolipid species dampened the activating effect of ORMs on ceramide synthase. The predictions pave the way for future guided experiments and have implications in engineering crops with higher biotic stress tolerance.

MCB 1818297 to EBC and RS and by the NSF EPSCoR Center for Root and Rhizobiome Innovation Grant 25-1215-0139-025 awarded to EC and RS at UNL. The funders had no role in study design, data collection and analysis, decision to publish, or preparation of the manuscript.

**Competing interests:** The authors have declared that no competing interests exist.

## Author summary

Due to their vital functional and regulatory roles in plant cells, increasing interest has gone into obtaining a complete understanding of the regulatory behavior of the sphingolipid pathway. However, the process of identifying new regulatory interactions is time consuming and often infeasible. To address this issue, ensemble modeling was used as an *in silico* method to test the ability of different regulatory schemes to predict all known pathway responses in a thermodynamically consistent manner. The analysis resulted in a significant reduction in the number of possible regulatory interactions. Mainly, the model predicts regulatory interactions between ceramides, ORMs, and ceramide synthases (especially class II). This framework can pave the way for biochemists to systematically identify plausible regulatory networks in understudied metabolic networks where knowledge on the underlying regulatory mechanisms is often missing. As future experimental works explore these predictions, an iterative cycle can begin wherein model predictions allow for targeted experiments which in turn generate results that can be reincorporated into the model to further increase prediction accuracy. Such a model-driven approach will significantly reduce the solution space traversed by the experimentalist.

## Introduction

Sphingolipids are a diverse group of membrane lipids essential in eukaryotic organisms. In plants, sphingolipids comprise up to 40% of the plasma membrane and are abundant components of cellular endomembranes such as the endoplasmic reticulum (ER), Golgi, and tonoplast [1,2]. Through their unique structural features, sphingolipids carry out several essential functions in plant cells. Glycosylated sphingolipids like glucosylceramides (GlcCer) and glycosylinositolphosphoceramides (GIPCs) contribute to membrane function and are involved in the trafficking of secretory proteins out of the cell [3,4]. The accumulation of other sphingolipids, namely long chain bases (LCBs) and ceramides, signal the initiation of programmed cell death (PCD) when plant cells are under environmental stresses, such as the presence of bacterial or fungal pathogens [5,6].

The sphingolipid biosynthesis pathway starts in the ER with the condensation of palmitoyl-CoA and serine to produce 3-ketosphinganine. This reaction is the first committed step towards sphingolipid LCB biosynthesis and is also the rate-limiting step of the sphingolipid biosynthetic pathway [7]. 3-Ketosphinganine is then reduced to sphinganine, which is the basic LCB. Sphinganine can then go through multiple modifications including unsaturation in Δ4 or Δ8 positions; hydroxylation at C4 and/or phosphorylation at its C1 position. LCBs can then be linked to a fatty acyl-CoA, typically with chain-lengths ranging from 16 to 26 carbon atoms, to produce ceramides [8,9] by the activities of ceramide synthases. In Arabidopsis, two classes of ceramide synthases were identified. Class I, encoded by Longevity Assurance Gene One Homolog2 (*LOH2*), mostly operates on acyl-CoAs of length 16 and dihydroxy LCBs and class II, encoded by *LOH1* and *LOH3*, act on acyl-CoAs containing more than 22 carbons, also referred to as very long chain fatty acids (VLCFAs) and tri-hydroxy LCBs [9,10]. The ceramide backbone is further modified by glycosylation at its C-1 position to form GlcCer or linked to inositol phosphate and further glycosylated in Golgi bodies to yield GIPCs, the most abundant glycosphingolipid in plant cells [1,11,12].

Similar to many other biochemical pathways, the first committed step catalyzed by serine palmitoyltransferase (SPT) constitutes the main regulatory point in the pathway [7]. Tight regulation on SPT ensures sufficient production of sphingolipid components to maintain cellular

growth, and simultaneously prevents the accumulation of PCD-inducing components under non-stress conditions. A small polypeptide of 56 amino acids referred to as the small subunit of SPT (ssSPT) is necessary for optimal activity of this enzyme [13]. Recently, orosomucoid-like proteins (or ORMs) have emerged as negative regulators of the sphingolipid pathway [14]. Studies conducted in *Arabidopsis thaliana* (hereafter Arabidopsis), *Saccharomyces cerevisiae* and mammalian cells have shown that the lack of functional ORM proteins results in accumulation of sphingolipids, especially ceramides and LCBs [14–17]. Interestingly, these proteins are essential to complete a life cycle in the model multicellular organisms Arabidopsis and mouse [16,17]. Although the exact regulatory mechanisms remain unknown, it has been shown that the ORM-SPT physical interaction is necessary to downregulate the activity of the enzyme [16,18].

Interestingly, in addition to the regulation at the first step of the biosynthetic pathway, ORMs were proposed to act as modulators of ceramide synthases [19]. It was shown that the overexpression of *ORM* genes leads to differential activity between the two classes of ceramide synthase. Activity of class I ceramide synthase (LOH2) was observed to decrease, while class II (LOH1/LOH3) activity was stimulated [19]. Conversely, downregulation of *ORM* gene expression produced the opposite effect. In addition, studies conducted on yeast and mammalian cells postulated a ceramide-ORM feedback regulation, in which the physical interaction between ceramides and ORMs leads to a decrease in SPT activity [20–22]. Taken together, these experimental observations started to uncover the global regulatory role played by ORMs in the sphingolipid pathway, however, there still remain critical knowledge gaps. For example, it is still not known whether ORMs have any regulatory interactions with the ceramide synthases and whether or not any other sphingolipid components interact with ORMs [19]. Furthermore, while several potential regulatory schemes were proposed to explain each of the observed pathway behavior, experimentally verifying each of the proposed mechanisms remains an impractical task.

Computational systems biology is a field concerned with modeling complex biological systems to test the feasibility of a predefined hypothesis or to generate new testable hypotheses based on existing observations [23]. Over the past decade, constraint-based genome-scale metabolic models (GSMMs) have been reconstructed and successfully used to probe the primary metabolism of several plant model systems including Arabidopsis [24–28]. The aim of such large-scale, multi-tissue models has been to (i) probe the activity of primary metabolism under different growth and environmental conditions [26], (ii) understand resource partitioning between source and sink tissues [27], and (ii) investigate shifts in central and energy metabolism associated with light and nutrient availability [26,28]. Therefore, detailed analysis of peripheral pathways such as the sphingolipid pathway is usually outside the scope of such studies and the pathways are either overly simplified [26,27] or omitted [28]. Furthermore, GSMMs rely on stoichiometric based analysis and therefore cannot capture relationships between flux, enzyme expression, metabolite concentration, and regulation [29].

A widely applied modeling framework that is used to address such limitations is kinetic modeling [30]. This framework describes the metabolic and regulatory processes occurring in the system through kinetic expressions (e.g. mass action, Michaelis-Menten). Starting with a set of initial conditions, the temporal behavior of the metabolic and regulatory network can be determined [31]. However, practical application of this framework is dependent on the availability of measured enzyme kinetic parameters for all enzymes in the network, which is usually not feasible. To overcome this limitation, ensemble kinetic modeling (EM) [32] was developed to sample through the entire allowable kinetic solution space to generate an ensemble of kinetic models that describe the system (see Materials and methods). This ensemble is then filtered using prior knowledge of the system's response to different genetic perturbations [33].

EM was used to capture the inherent non-linearity of metabolic systems and to identify metabolic bottlenecks in the production of various industrially relevant compounds [34]. In addition, EM was used to predict the presence of regulatory interactions occurring in biochemical pathways [35]. Implementation of such procedure can therefore greatly accelerate the process of hypothesis generation and testing by predicting plausible regulatory mechanisms that satisfy the observed experimental behavior of the pathway under different growth conditions. This analysis can subsequently be followed up with experimental testing for validation and iterative model curation.

A number of studies have used different forms of kinetic modeling to study sphingolipid metabolism in yeast and mammalian cells [36–39]. These studies used different experimental data sets such as lipidomics, transcriptomics, and fluxomics to parametrize a predefined kinetic model describing the sphingolipid pathway. In most cases, the implemented methods assumed that the sphingolipid regulatory network was completely understood [36–38]. Furthermore, they did not account for the possibility that more than one parameter set can satisfy the observed experimental measurements. A recent study aimed at analyzing sphingolipid metabolism in yeast addressed these issues by implementing a new framework called inverse metabolic control analysis (IMCA) [39]. The authors used this framework to identify the key enzymes responsible for the observed sphingolipid profile. However, the study did not incorporate any regulatory proteins such as ORMs into their analysis.

In this work, a kinetic model of the Arabidopsis sphingolipid pathway was constructed to predict the pathway's response to various internal and external perturbations. By using the EM approach, we identified a highly plausible regulatory interaction between ORM proteins and different components of the sphingolipid network. These interactions were deemed plausible as their presence was required in order to obtain the experimentally observed behavior of the system. By testing different sets of regulatory mechanisms, the implemented framework was used to eliminate postulated interactions which were not consistent with the experimentally observed response of the network to genetic perturbations. The novel prediction made by the analysis is the presence of a ceramide-ORM-ceramide synthase (class II) regulatory interaction. In this scheme, ORMs activate class II ceramide synthases (LOH1/LOH3); however, upon ceramides accumulation ORMs are repressed and not able to activate CS II causing a buildup of LCBs. These results illustrate how kinetic models can be used to predict regulatory mechanisms in biochemical systems despite lack of prior knowledge on enzyme kinetics. These predictions will result in a metabolic and regulatory model of the sphingolipid network capable of more accurately predicting the pathway's response to different genetic manipulations as well as its response to biotic and abiotic stresses.

## Results

### Metabolic network reconstruction

The reconstructed network comprises a knowledgebase of all the active metabolic transformations involved in sphingolipid biosynthesis in Arabidopsis (see S1 File). The network comprises 77 reactions associated with 24 genes. This resulted in an elementary reaction network consisting of 374 elementary reactions, excluding the elementary steps incorporated to describe regulatory mechanisms (see Materials and methods). The large difference between the number of reactions and number of genes is due to the enzyme promiscuity present in the sphingolipid synthesis pathway. In Arabidopsis, the activity of the Δ4 desaturase is minimal in most tissue types [40], therefore this reaction was not included into the network. Moreover, the majority of acyl-CoAs found in sphingolipids are either C16 or C24 fatty acids [8,41]. Hence, the relatively negligible concentrations of other chain-length fatty acids were combined

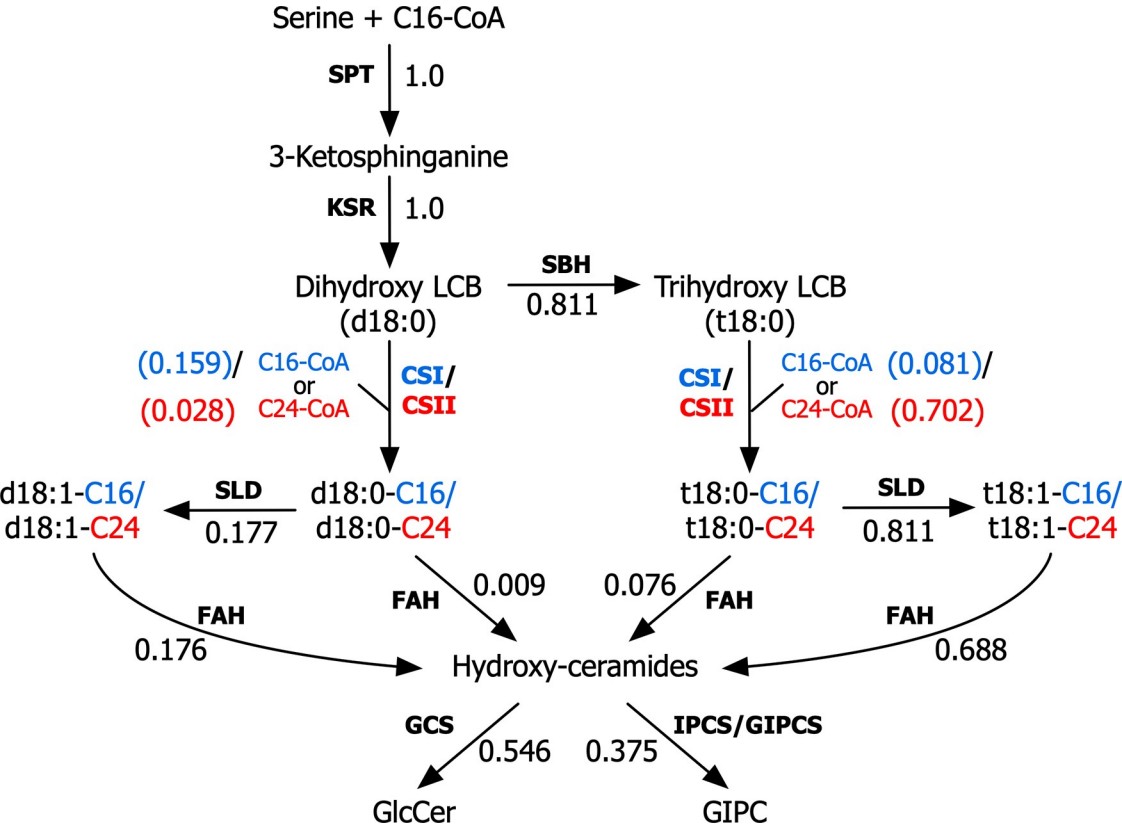

**Fig 1. Simplified depiction of the sphingolipid pathway.** A metabolic network displaying the main reactions constituting the sphingolipid pathway. Blue flux values correspond to reactions catalyzed by CSI and red flux values correspond to reactions catalyzed by CS II. SPT: serine palmitoyltransferase, KSR: 3-ketosphinganine reductase, SBH: LCB C-4 hydroxylase, CSI: class I ceramide synthase, CSII: class II ceramide synthase, SLD: LCB Δ8 desaturase, FAH: fatty acid hydroxylase, GCS: glucosylceramide synthase, GlcCer: glucosylceramide, GIPCS: glycosyl inositolphosphoceramide synthase, LCB: long chain base. C16 and C24 are fatty acids of length 16 and 24 carbons, respectively.

with one of these two groups based on their chain length. This was done to reduce redundancy in the added reactions and to avoid having an exaggerated number of kinetic parameters, which could potentially lead to overfitting.

Fig 1 shows the metabolic map used to construct the kinetic model. Several assumptions had to be made on the order of modifications occurring to the different components in the sphingolipid pathway. It was assumed that the enzyme LCB C4 hydroxylase (SBH1 and SBH2) mainly hydroxylates free long chain bases instead of ceramides containing dihydroxylated LCBs. This was based on previous evidence which showed that the *sbh1 sbh2* double mutant accumulated sphingolipids enriched in C16 fatty acids with dihydroxy LCBs coming from the ceramide synthase class I [42]. Moreover, based on the findings reported by Konig et al. [43], it was assumed that the acyl-hydroxylating enzyme fatty acid α-hydroxylase (FAH1 and FAH2) converted only ceramide-bound fatty acids and not free fatty acids. LCB Δ8 desaturase (SLD1 and SLD2) was also assumed to produce unsaturated LCBs bound to ceramides and not free LCBs [44,45]. Again, this was done to avoid redundancy, as omitting these assumptions would have led to an exponential increase in the number of reactions and therefore fitting parameters. In addition, it was assumed that sphingolipid turnover rates were negligible compared to their rate of formation. This assumption was necessary to calculate reference steady-state fluxes. The linear GIPC forming pathway involving IPC-synthases and several glycosyl or

glucuronyl-transferases [46] was lumped into one step, as no information was present on the relative kinetics of the participating enzymes. Finally, since the majority of the glycosylated sphingolipids (GlcCers and GIPCs) contain ceramides with hydroxylated fatty acids, no reactions were added to convert non-hydroxylated ceramides into these complex forms.

## Generation of kinetic parameters

The Ensemble modeling framework requires the input of both the reference steady-state fluxes and the standard Gibbs free energies of the modeled reactions to generate the initial set of kinetic parameters [32] (see Fig 2). The fluxes were determined by measuring the growth rate and sphingolipid profiles of Arabidopsis seedlings at multiple time points during exponential growth (see Materials and methods). By assuming negligible sphingolipid turnover and a constant sphingolipid composition, measurements of growth and composition were used to determine the accumulation rate of each sphingolipid component. Flux balance analysis (FBA) [47] was subsequently used to calculate the fluxes of the internal reactions in the network (see Materials and methods). The values displayed in Fig 1 are normalized to SPT to illustrate how the flux going through this reaction is branched into the different sphingolipid products. As can be seen, the majority of the produced LCBs (73%) are channeled through class II ceramide synthase to produce VLCFA containing ceramides. Furthermore, under normal conditions, practically all of the sphingolipid components are non-phosphorylated (<1% phosphorylation). The activity of the complex sphingolipid producing enzymes GCS and GIPCS were similar. However, whereas the majority of GIPC products were VLCFA-containing ceramides, 30% of the flux through GCS went towards synthesizing C16-containing ceramides. This was an interesting observation for the wild type, since previous CSI knockout studies found that elimination of the C16-ceramide producing ceramide synthase has no observed change in phenotype under normal growth conditions [8].

The component contribution method [48] was subsequently used to calculate the standard Gibbs free energy of each reaction in the pathway. As can be seen from the values in Table 1, the standard free energy for most of the biochemical transformations in the pathway are negative and relatively far from equilibrium. Therefore, these reactions have a wide range of substrate/product concentration ratios that can result in an overall forward flux of the reaction [49] (see S2 File). However, the first and rate limiting step of the pathway, the reaction catalyzed by SPT [1], has a slightly positive standard free energy. This means that a positive substrate to product concentration ratio needs to be maintained in order for the reaction rate to stay in the forward direction. Furthermore, the rate of the reaction is directly affected by any changes in the actual free energy, and therefore in the concentrations of the participating reactants [49]. Once the required inputs were determined, a MATLAB implementation of the ensemble modeling framework was customized to generate the kinetic parameters (see Materials and methods). This process was repeated for each of the tested regulatory schemes. Fig 3 illustrates how the generated set of kinetic parameters result in the measured steady-state fluxes for the reference (wild type) steady-state and how the generated models are screened based on their response to different perturbations.

## The effect of perturbations on the generated model

To determine the feasibility of the metabolic network without any additional regulatory interactions, we introduced perturbations in the form of enzyme overexpression or repression and determined the predicted response of each model in the ensemble to the introduced perturbation. The predicted response was then compared with the experimentally observed response which was imposed as a 'filtration step'. Table 2 shows the different perturbations introduced

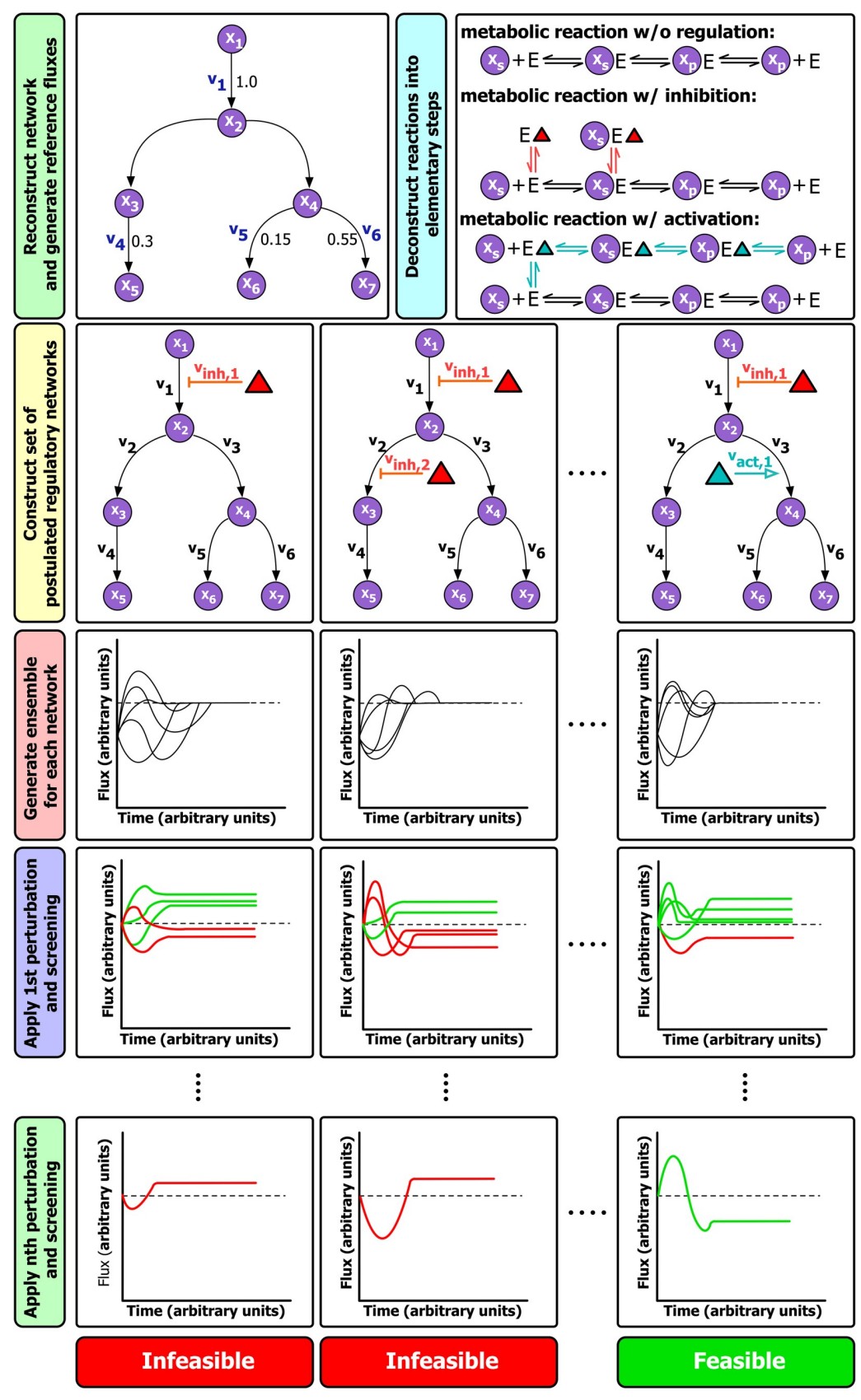

**Fig 2. Workflow for using Ensemble modeling to predict regulatory mechanisms.** Circles: metabolites, triangles: regulators (red and blue signify inhibitors and activators, respectively). Dotted lines represent the steady-state flux. Green lines indicate flux profiles corresponding to models that pass the applied perturbation and red lines signify models that did not pass. $X_S$: substrate, $X_P$: product, $V_{inh}$: inhibition reaction flux, $V_{act}$: activation reaction flux.

and the applied filters linked to each of them. One of the interesting observations made prior to this analysis was the differential activity in the two classes of ceramide synthase resulting from a perturbation to the level of ORMs in the system. Previous studies postulated that this differential activity potentially points to a possible regulatory interaction between ORMs and the ceramide synthases [19]. However, it was found that a large fraction of the generated models displayed this behavior without the need for any regulatory intervention, meaning that this behavior is more likely to emerge from the kinetic properties of the two enzyme classes. Furthermore, it was observed that none of the models had satisfied the filtration step requiring an increase in the concentration of the LCB sphinganine (d18:0) compared to the wild type during the overexpression of class I ceramide synthase. This observation is expected since overexpressing ceramide synthase should intuitively decrease the concentration of LCBs which are considered substrates to this enzyme. In addition, none of the models satisfied the experimental observation that the concentration of the VLCFA containing ceramide d18:1-hC24 increases during ORM overexpression. This was due to a decrease in the overall amount of LCBs produced resulting from the reduced rate through SPT. The absence of models passing these two filtration steps (see Table 2 for details) indicated that the incorporation of additional regulatory interactions was required for the model to satisfy the available experimental observations.

## Predicting the regulatory scheme of the sphingolipid pathway

To predict plausible regulatory schemes capable of reproducing all of the observed responses, we started with a set of 23 schemes that had been postulated in previous works [19,22]. Recent work in mammalian cells and yeast had found that accumulation of ceramides had an inhibitory effect on SPT activity, and that this interaction had been facilitated through ORMs [22]. In addition, previous work on the role of ORMs on the sphingolipid pathway in plants had hypothesized the possibility of a regulatory interaction between ORMs and ceramide synthases [19]. Therefore, the set of starting regulatory networks to be tested included different combinations of regulatory interactions between ORMs and the ceramide synthases and/or ceramides, as well as ceramide inhibition of SPT. These interactions were either in the form of an enzyme activation or inhibition by a component in the metabolic network. S3 File has a description of how the 23 schemes were generated and an explanation of how regulatory interactions were incorporated into the network.

**Table 1. Standard Gibbs free energy values for reactions in the sphingolipid pathway.**

| Reaction | Reaction Name | $\Delta G'^0$ (kJ/mol) |
|---|---|---|
| SPT | serine palmitoyltransferase | 8.9 |
| KSR | 3-ketosphinganine reductase | -17.1 |
| SBH | LCB C-4 hydroxylase | -397.4 |
| CS1 | class I ceramide synthase | -25.5 |
| CS2 | class II ceramide synthase | -27.6 |
| FAH | fatty acid hydroxylase | -93.0 |
| GCS | glucosylceramide synthase | -4.5 |
| GIPCS | glycosyl inositolphosphoceramide synthase | -5.1 |

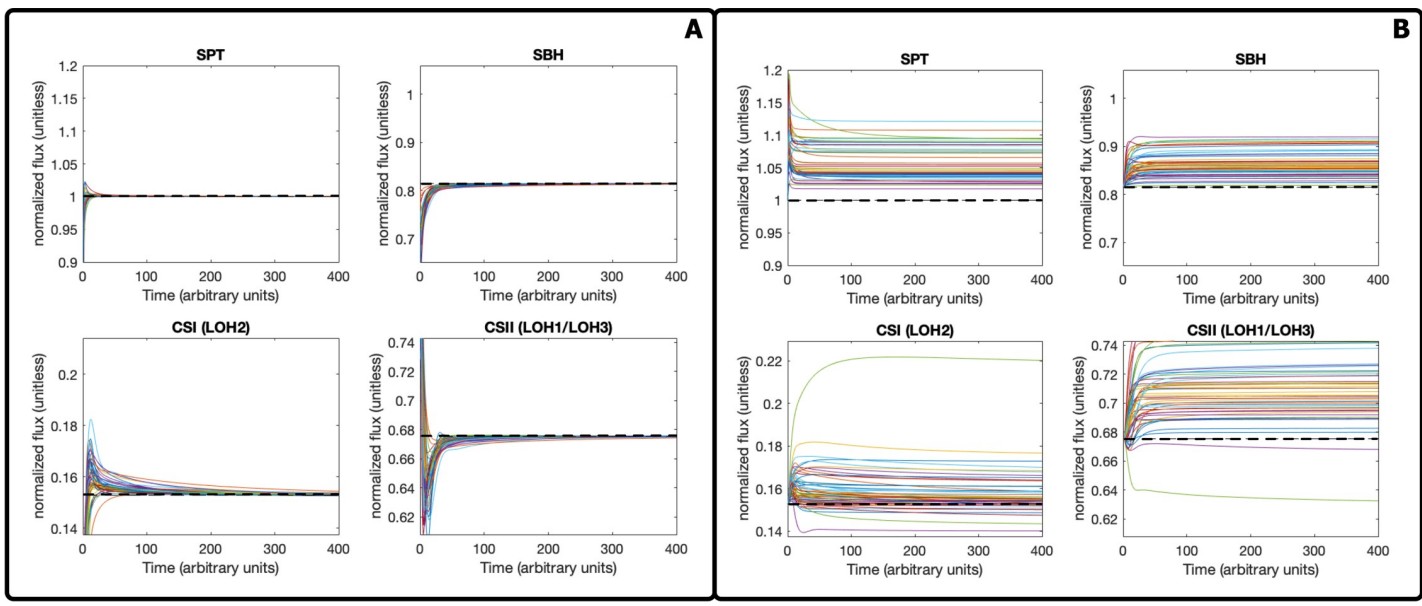

**Fig 3. Model behavior under different conditions.** (A) The flux profiles of different reactions during the reference (wild type) state. All of the models in the ensemble reach the same steady-state flux. The dashed line refers to the reference steady-state value. (B) The flux profiles of different reactions during a perturbed state. The displayed perturbation is an ORM repression. The dashed line refers to the reference steady-state value.

The majority of the tested schemes resulted in no change to the number of filters passed compared with the starting metabolic network. The two filters discussed in the previous section had remained problematic. However, it was found that the schemes including ceramide repression of ORMs were able to satisfy all of the observed responses. It is noted that although the model captures the repression of ceramides on ORMs as a decrease in the concentration of these proteins, the observed behavior would be the same if instead ceramides repressed ORM's binding efficiency to the regulated enzymes. Out of the 23 starting schemes, three regulatory networks all requiring ceramide repression of ORMs had passed the implemented filtration steps and were therefore considered candidates for further experiments to validate the presence of a regulatory interaction. Furthermore, it was found that the presence of an activating interaction between ORMs and CSII produced a larger number of models satisfying the observed responses. Fig 4 below displays this hypothesized regulatory network.

**Table 2. List of perturbations and associated filtration steps used to screen the ensemble of models.**

| Perturbation | Applied Filters |
|---|---|
| ORM RNAi[a] | ↑ CSI activity |
| | ↓ CSII activity |
| **ORM OE** | ↓ CSI activity |
| | ↑ CSII activity |
| | ↓ t18:0 and d18:1-hC16 concentration |
| | ↑ d18:1-hC24 concentration |
| **CSI OE** | ↑ d18:0 concentration |
| | ↑ "C16-cer" concentration |
| | ↓ "C24-cer" concentration |

OE refers to overexpression and is modeled through an increase in the corresponding enzyme.

a RNAi refers to inhibition of the RNA coding for ORM expression.

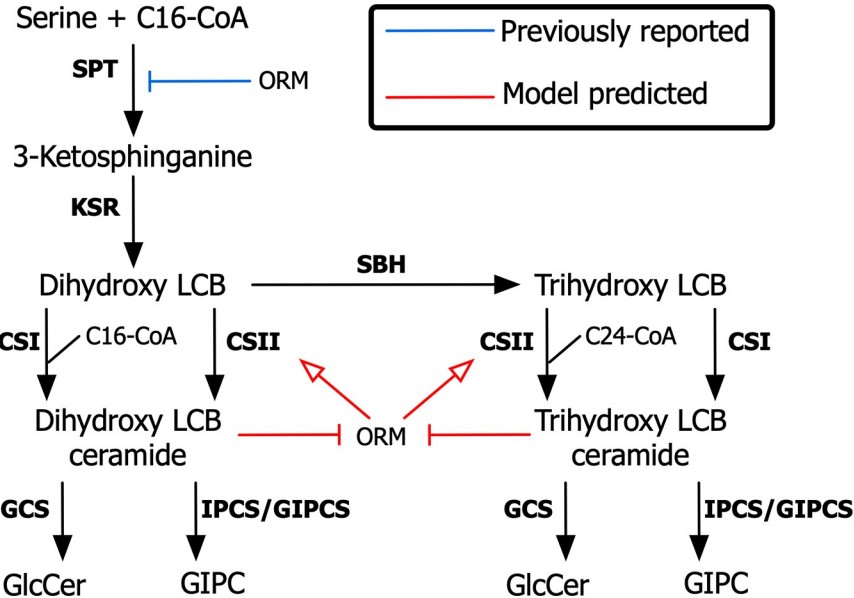

**Fig 4. Predicted regulatory scheme of the sphingolipid pathway.** A metabolic and regulatory network displaying the mechanism predicted to be responsible for the emerging observed behavior of the pathway. Ceramides repress ORM-mediated activation of CSII. Blue lines indicate regulatory interactions identified in previous studies and red lines indicate novel interactions predicted in this study.

As can be seen from Fig 4, the role of both hypothesized regulatory mechanisms is to cause an accumulation in dihydroxy-LCBs (d18:0) during CSI overexpression. The increase in ceramides resulting from the overexpression of LOH2 ceramide synthase (CSI) results in the repression of ORM-mediated activation of CSII. Analysis of the passing models showed that this does not cause a significant change in the flux through SPT. The main effect of this regulation is to dampen the activating role of ORMs on CSII. This causes the concentration of LCBs to increase, satisfying the imposed filter.

To gain further insight into what factors allowed a subset of the ensemble of kinetic parameters associated with the regulatory network (Fig 4) to satisfy the experimental observation, those parameters were compared with the ones that did not pass the applied filtrations steps. Due to the complexity in the relationship between the kinetic parameters of any given model, it was difficult to pinpoint any differences between the two sets by comparing elementary kinetic parameters. Therefore, these parameters were lumped in order to calculate apparent kinetic parameters [49] (see S4 File). Subsequently, a two-sample Kolmogorov-Smirnov test (KS test) [50] was performed to determine whether there was any statistical significance in the distributions of parameters that passed the applied filtration steps compared to those that did not. Interestingly, it was found that there was a significant (p < 0.001) difference for three of the lumped parameters. Namely, both the maximal catalytic efficiency ($k_{cat}^+$) and substrate associated Michaelis constant ($K_s$) associated with LCB C4-hydrolase (SBH) were significantly different between models that passed and those that did not (S4 File). This points to the essential role that SBH plays in branching di- and trihydroxy LCBs. Furthermore, the distribution of $k_{cat}^+$ values associated with the activated CSII pathway also displayed significant differences between the two subsets of models (passed vs. failed). This analysis indicates that LCB branching at SBH, and the binding of ORMs to CSII may play a vital role in maintaining homeostasis within the sphingolipid pathway.

## Discussion

Sphingolipids constitute a major class of lipids in plants and play several functional and regulatory roles in plant cells [51]. The regulatory mechanisms governing the activity of the sphingolipid biosynthesis pathway are not completely understood. Specifically, even though it is well documented that ORM proteins negatively regulate SPT, questions regarding the role of these proteins in regulating different enzymes in the pathway remain unanswered [15,19,22]. In this work, we used a model guided approach to predict a regulatory scheme consistent with the pathway's observed behavior under different conditions, the up- and downregulation of *ORM* gene expression and the overexpression of ceramide synthase class I (LOH2). The predictions made pave the way for future guided experiments and have implications in engineering crops with higher biotic stress tolerance.

Previous reports of computational analysis of sphingolipid metabolism have focused on modeling the pathway's activity in yeast and mammalian cells under different contexts [36–39]. However, most of the applied frameworks either did not consider regulatory interactions or assumed a pre-defined regulatory network. A recent study of sphingolipid metabolism in yeast developed a framework to determine the regulatory effect each enzyme had on the pathway but did not incorporate the role of regulatory proteins such as ORMs [39].

In this study, a kinetic model comprised of elementary reactions describing both metabolic transformations and regulatory interactions in the sphingolipid pathway was constructed. To avoid both redundancy and overfitting, several assumptions were made to simplify the metabolic network. First, the order of reactions was assumed to be fixed based on evidence from previous works. It is possible that some of the enzymes are promiscuous in the order in which they catalyze a given reaction (e.g., LCB modification). Second, the concentration of cofactors and palmitoyl-CoA were considered to be constant both in the reference (wild type) and perturbed model. These metabolites participate in a large number of reactions in the cell and were therefore assumed to be tightly regulated (since a change in their concentration will have an effect on all the reactions they participate in). The addition of this constraint greatly reduced the number of fitting parameters required, as the cofactors are modeled to be a part of the enzyme complex. In addition, diffusion and transport rate limitations of sphingolipids entering and exiting the ER were not considered. This assumption was necessary as the transport kinetics of the pathway are not known. Finally, it was assumed that sphingolipid turnover rates were negligible compared to their rate of formation. This assumption was necessary to calculate reference steady-state fluxes. Labeling studies using the stable isotope nitrogen 15 will be conducted to determine rates of turnover reactions and these results will be used to further improve the model.

The constructed network was used to generate a set of kinetic parameters that converge to the same reference steady-state fluxes. By incorporating data on how the pathway responds to the different perturbations represented in Table 2, different regulatory schemes were tested to determine which ones were consistent with the observed data. The experimentally observed pathway responses to different perturbations (enzyme over-expression (OE)/downregulation) were incorporated into the framework as filtration steps that the generated models needed to satisfy. This was implemented using a modified ensemble modeling framework (see Materials and methods). Namely, the process of model screening was automated through the addition of a new module which filters through the initial ensemble of kinetic parameters. Furthermore, the incorporation of an activating regulatory mechanism was amended to ensure the generation of models consistent with the reference steady-state.

Starting with only the known regulatory interaction between ORMs and SPT, it was found that this mechanism alone was sufficient to obtain the observed differential activity of the two

classes of ceramide synthases during ORM OE or repression. This observation was counterintuitive as both enzymes use LCBs as their substrate. The model showed that this behavior could be explained solely based on the kinetics of the two classes of enzymes and does not require any further regulation of either ceramide synthase. However, this scheme was still incomplete, as it did not satisfy other observed behaviors in the pathway. Namely, the observation that LCBs accumulate during *LOH2* (CSI) OE required additional regulatory mechanisms to be incorporated into the network. By testing a set of plausible regulatory schemes, it was observed that the addition of a ceramide-ORM inhibitory interaction and an ORM activation of class II ceramide synthases was required for all observed filtration steps to be satisfied. This scheme is depicted in Fig 4. In this scenario, the prediction of the model indicates two layers of regulation under normal conditions, (1) ORMs activate the synthesis of C24-containing ceramides by LOH1/LOH3 (CSII) and (2) the accumulation of ceramides represses this ORM-mediated activation on CSII. The first prediction is consistent with a previous study where overexpression of ORMs resulted in increased activity of class II CS that preferentially use VLCFA and trihydroxy LCBs as substrates [19]. However, the effect of the accumulation of ceramides in this regulation has not been reported. This repression by ceramides could also be explained by a competitive inhibition of ceramide synthase.

In mammalian cells, it has been shown that ceramides, or downstream metabolites, are involved in regulating the expression of ORM proteins, ceramide accumulation leads to increased ORM protein levels [52]. In addition, ceramides mediate SPT inhibition by ORMs [22]. The role of ORMDL proteins in the regulation of ceramide synthesis was tested using mammalian cell cultures. However, in this system ORMDLs do not regulate ceramide synthases [20]. The biological implications of the ceramide-ORM-CSII regulation depicted in the present study could be specific to plants where ceramides containing VLCFA and trihydroxy LCBs are very abundant compared to mammalian cells. This regulation could be relevant under bacterial and fungal infections where differential accumulation of C16-containing ceramides and VLCFA-containing ceramides might be related to the defense response. Further studies will be conducted with the fungal toxin fumonisin B1 (FB1), a ceramide synthase inhibitor that acts preferentially on LOH1 [53] promoting the accumulation of C16 ceramides through LOH2. These results will be compared with the LOH2 OE scheme presented in this work. In addition, future experimental work will focus on verifying the interaction of ORMs-LOHs and the ORM-mediated activation of CSII. Specifically, this will be achieved through co-immunoprecipitation (co-IP) experiments that will be used to determine the presence of physical interactions between ORM1/2 proteins and ceramide synthases (LOH1/2/3).

The main strength of using the ensemble modeling framework to approach such challenges is its ability to eliminate regulatory schemes which appear to be plausible on first sight. As demonstrated in this work, the analysis significantly reduced the number of plausible regulatory schemes to be tested experimentally. Furthermore, due to the computational intractability of testing all combinations of regulatory interactions in the pathway, it is possible that other regulatory networks can also satisfy all filtration steps applied in this study. For example, ORMs (or other components) could also be involved in regulating other downstream enzymes like GCS. As more experiments are carried out on different Arabidopsis mutants and transgenic lines, the number of filtration steps will increase, potentially leading to fewer schemes that satisfy all observations. Furthermore, experiments aimed at looking for regulatory interactions between ORMs and different parts of the network might unravel additional interactions that were not tested in this work. These schemes can easily be incorporated into the model as the metabolic reactions have already been constructed. This highlights the need for future experimental work to test the predictions of this study, which will allow the model to be updated in a more systematic manner. We emphasize the importance of such model-driven

approaches in addressing such problems as the one in this study, where the aim of the predictions is (i) to reduce the solution space that the experimentalist has to cover and (ii) to pinpoint parts of the pathway where more data is needed to obtain a more accurate model. In this regard, this work can be considered as the first step in a continuously updated design-test-refine cycle aimed at obtaining an accurate mechanistic understanding of the behavior of the sphingolipid pathway in plants.

## Materials and methods

### Sphingolipid compositions

The concentrations of LCB, ceramides, GlcCer, and GIPC in 12 to 15-day-old wild-type (Columbia-0) seedlings were obtained from a previous study [16].

### Growth rate

Arabidopsis seedlings were grown on Murashige and Skoog (MS) medium supplemented with 1% sucrose and 0.8% agar (pH 5.7) with 16 h light ($100\mu mol/ m^{-2} s^{-1}$) 8 h dark conditions at 22˚C and 55% humidity. Three replicates, consisting of ten seedlings, were sampled at each time point (5 d, 10 d, 15d and 20 d). The seedlings were freeze-dried, and the dry weight recorded. Subsequently, a semi-log plot of the data was constructed, and linear regression was used to obtain a best-fit line, with a slope corresponding to the growth rate (see S1 File).

### Ensemble modeling

EM describes both the metabolic and regulatory state of a pathway as a set of elementary steps obeying mass-action kinetics. This allows the construction of multiple sets of regulatory schemes, which can be subsequently screened to determine the thermodynamic and kinetic feasibility of each network. This section contains a brief description of how the set of kinetic parameters can be obtained for a unimolecular enzymatic reaction with no regulation. Complex reaction schemes follow the same general procedure but contain more elementary steps.

### Determining steady-state flux distributions

Parsimonious Flux Balance Analysis (pFBA) [54] was used to generate the steady state flux distribution. pFBA is analogous to FBA but adds an outer objective that minimizes the sum of all reaction fluxes. Objective tilting [55] was used to formulate both objectives in one function as shown below.

$$Maximize \ v_{biomass} - 0.0001 \sum_{j \in J - v_{biomass}} |v_j|$$

$$subject \ to$$

$$\sum_{j \in J} S_{ij} \cdot v_j = 0 \forall i \in I \tag{1}$$

$$LB_j \leq v_j \leq UB_j \forall j \in J \tag{2}$$

Where $I$ and $J$ are the sets of metabolites and reactions in the model, respectively. $S_{ij}$ is the stoichiometric coefficient of metabolite $i$ in reaction $j$ and $v_j$ is the flux value of reaction $j$. Parameters $LB_j$ and $UB_j$ denote the minimum and maximum allowable fluxes for reaction $j$, respectively. $v_{biomass}$ is the flux of the biomass reaction which mimics the cellular growth rate.

The experimentally measured cellular composition of each of the sphingolipid components, obtained from a prior study [16], was used to establish the stoichiometric coefficients of the biomass equation. Since the majority of ceramides and complex sphingolipids contained either 16 or 24 carbon fatty acids, other acyl-CoA lengths were pooled together with one of the two groups depending on whether they contained more or less than twenty carbons. The growth rate was also determined experimentally to constrain the rate of the biomass reaction. Subsequently, by assuming negligible sphingolipid turnover rates, pFBA was used to calculate the reaction rates for each enzymatic reaction in the network.

## Generating initial ensemble of models

The elementary steps for a unimolecular enzymatic reaction can be written as:

$$X_1 + E \underset{V_{i,2}}{\overset{V_{i,1}}{\leftrightarrow}} X_1 E \underset{V_{i,4}}{\overset{V_{i,3}}{\leftrightarrow}} X_2 E \underset{V_{i,6}}{\overset{V_{i,5}}{\leftrightarrow}} X_2 + E$$

where, enzyme $E_i$ converts metabolite $X_i$ into $X_{i+1}$. Each step is reversible and has a rate of $v_{i,2j-1}$ in the forward direction and $v_{i,2j}$ in the reverse direction. Since each elementary step follows mass-action kinetics, the reaction rate can be expressed as the product of the reactant concentrations and a rate constant. Therefore, the rate of the forward reaction of the first step can be written as:

$$v_{i,1} = k_{i,1}[X_i][E_i]$$

where, $k_{i,1}$ is the rate constant associated with the forward reaction of the first step. This equation is subsequently normalized by the reference steady-state metabolite concentration and the total enzyme concentration to yield:

$$v_{i,1} = \left(k_{i,1} E_{i,total}{}^{ref} X_i^{ss,ref}\right) \frac{[X_i]}{X_i^{ss,ref}} \frac{[E_i]}{E_{i,total}{}^{ref}} = \widetilde{K_{i,1}^{ref}} \tilde{X}_i \widetilde{e_{i,1}}$$

where, $E^{ref}_{i,total}$ is the total enzyme concentration and $X_i^{ss,ref}$ is the steady state metabolite concentration, both at the reference steady state. Preforming this normalization alleviates the need to measure metabolite and enzyme concentrations. This equation is subsequently converted to the log-linear form:

$$\ln v_{i,1} = \ln \widetilde{K_{i,1}^{ref}} + \ln \tilde{X}_i + \ln \widetilde{e_{i,1}}$$

During steady-state, $X_i$ becomes 1, therefore the $\ln(X_i)$ term can be omitted. This yields:

$$\ln v_{i,1} = \ln \widetilde{K_{i,1}^{ref}} + \ln \widetilde{e_{i,1}}$$

Next, the reversibility of each reaction step ($R_{i,j}$) is sampled in order to determine the rate of each elementary reaction $v_{i,j}$

$$R_{i,j} = \frac{\min(v_{i,2j-1}, v_{i,2j})}{\max(v_{i,2j-1}, v_{i,2j})}$$

$R_{i,j}$ can take on any value between 0 and 1 (where 0 is an irreversible reaction step) and is further constrained by the Gibbs free energy of the reaction as follows

$$\left(\frac{\Delta G_i}{RT}\right)_{LB} \leq sign\left(V_{i,net}^{ref}\right) \sum_j \ln R_{i,j}^{ref} \leq \left(\frac{\Delta G_i}{RT}\right)_{UB}$$

The reaction rates for each step ($v_{i,j}$) can then be determined from the net enzymatic reaction rate ($V^{ref}_{i,net}$)

$$v^{ref}_{i,2j-1} - v^{ref}_{i,2j} = V^{ref}_{i,net}$$

Finally, the enzyme fractions for each elementary step ($e_{i,j}$) are sampled between 0 and 1 in order to calculate the kinetic parameter ($K_{i,j}{}^{ref}$). An additional constraint is imposed to ensure that the sum of enzyme fractions for each reaction is unity.

$$\sum_{j=1}^{n_j} \widetilde{e^{ref}_{i,j}} = 1$$

This procedure is repeated up to 10,000 times to generate an ensemble of kinetic parameters that all reach the predefined reference steady-state. Next, a system of ordinary differential equations describing the metabolic network is solved to obtain the net steady-state fluxes.

$$\frac{d\widetilde{X}_i}{dt} = \frac{1}{X^{ss,ref}_i} \left( \sum v_{generation} - \sum v_{consumption} \right)$$

$$\frac{d\widetilde{e}_i}{dt} = \frac{1}{E^{ref}_{i,total}} \left( \sum v_{generation} - \sum v_{consumption} \right)$$

A modified version of the MATLAB implementation developed by Tran et. al. [32] was used to carry out the simulations in this work. The kinetic model and all scripts required to generate the results in this study can be found at the link below.

https://github.com/aalsiyabi/plant_sphingolipid_kinetic_model.git

## Model screening

After the initial ensemble of models are generated, a number of enzymatic perturbations are introduced to analyze each model's response. The perturbations take on the form of enzyme overexpression, inhibition, or knockout and are formulated as follows

$$v_{i,1} = E_{i,r} \widetilde{K^{ref}_{i,1}} \widetilde{X}_i \widetilde{e_{i,1}}$$

Where $E_{i,r}$ represents the introduced fold change in enzyme expression compared to the reference state. Although the original MATLAB implementation [32] incorporated both inhibition and activation regulatory mechanisms, it was observed that no models were generated when activation was incorporated into the regulatory network. Therefore, modifications were implemented to ensure that the mass balances of the introduced activator-enzyme complexes are satisfied. Furthermore, the fluxes of the activated pathway(s) were integrated with the flux going through the main reaction to ensure an accurate response to changes in the concentration of the activator.

After each perturbation, the predicted enzymatic response is compared with the experimentally observed behavior. Models resulting in contradictory enzymatic behavior are disregarded. In this manner, the initial set of models is filtered through multiple rounds of perturbations, until only a small set of highly predictive models remain. A new module was added to the original implementation to automate the screening process. Perturbations with experimentally known responses were incorporated into the module in order to automatically screen the initial ensemble of models. The output is a subset of models passing all filtration steps. It is noted that the order in which the screenings are introduced does not affect the final outcome.

### Constructing multiple sets of possible regulatory networks

As described earlier, regulatory events can be described as elementary reaction steps [56]. The regulatory network is input as a stoichiometric matrix (Sreg) similar in size to the S matrix used to describe the metabolic network. The element $Sreg_{i,j}$ describes the type of regulation metabolite i confers on enzyme j, and can either be activating or inhibitory. Therefore, by constructing multiple Sreg matrices corresponding to different possible regulatory schemes, the EM approach can be applied on each regulatory network individually. An initial ensemble of models converging to the experimentally determined reference steady-state is then constructed as detailed previously. During this step, kinetic parameters associated with regulatory processes and those associated with metabolic processes are sampled simultaneously. Furthermore, after generating and filtering the set of kinetic parameters for each network, the regulatory scheme passing all the screening steps is hypothesized to be the most accurately representative of the biological system. A list of the tested regulatory networks is provided in S3 File.

## Supporting information

**S1 File. Sphingolipid biosynthesis metabolic network.**
(XLSX)

**S2 File. Standard Gibbs free energy of reactions in the network.**
(DOCX)

**S3 File. Set of regulatory schemes tested in this work.**
(DOCX)

**S4 File. Calculating apparent kinetic parameters and KS scores.**
(DOCX)

## Acknowledgments

We thank Dr. Jennifer Markham, Dr. Gongshe Han, and Dr. Teresa M. Dunn for their valuable input.

## Author Contributions

**Conceptualization:** Adil Alsiyabi, Ariadna Gonzalez Solis, Edgar B. Cahoon, Rajib Saha.

**Data curation:** Adil Alsiyabi, Ariadna Gonzalez Solis.

**Formal analysis:** Adil Alsiyabi, Ariadna Gonzalez Solis.

**Funding acquisition:** Edgar B. Cahoon, Rajib Saha.

**Investigation:** Adil Alsiyabi, Ariadna Gonzalez Solis.

**Methodology:** Adil Alsiyabi.

**Project administration:** Rajib Saha.

**Resources:** Edgar B. Cahoon, Rajib Saha.

**Software:** Adil Alsiyabi, Rajib Saha.

**Supervision:** Edgar B. Cahoon, Rajib Saha.

**Validation:** Adil Alsiyabi.

**Visualization:** Adil Alsiyabi.

**Writing – original draft:** Adil Alsiyabi, Ariadna Gonzalez Solis.

**Writing – review & editing:** Adil Alsiyabi, Ariadna Gonzalez Solis, Edgar B. Cahoon, Rajib Saha.

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
