## [Decision Letter · Decision Letter 0]

19 Oct 2020

Dear Dr. Saha,

Thank you very much for submitting your manuscript "Dissecting the regulatory roles of ORM proteins in the sphingolipid pathway of plants" for consideration at PLOS Computational Biology.

As with all papers reviewed by the journal, your manuscript was reviewed by members of the editorial board and by several independent reviewers. In light of the reviews (below this email), we would like to invite the resubmission of a significantly-revised version that takes into account the reviewers' comments.

We cannot make any decision about publication until we have seen the revised manuscript and your response to the reviewers' comments. Your revised manuscript is also likely to be sent to reviewers for further evaluation.

Sincerely,

Kourosh Salehi-Ashtiani

Guest Editor

PLOS Computational Biology

Jason Papin

Editor-in-Chief

PLOS Computational Biology

Reviewer's Responses to Questions

**Comments to the Authors:**

Reviewer #1: In the manuscript entitled “Dissecting the regulatory roles of ORM proteins in the sphingolipid pathway of plants”, the authors used an Ensemble modeling approach to narrow down possible regulatory mechanisms of the sphingolipid pathway in Arabidopsis seedlings. They experimentally determined the growth rate and reconstructed the sphingolipid pathway, including both metabolic and regulatory processes. The sets of generated kinetic parameters converged to steady-state fluxes, which were used as reference for the perturbed Ensemble models. The models were screened against published observations by applying a range of perturbations (e.g., over- and under-expression of ceramide synthases) and associated filters. Finally, using 23 schemes of potential regulatory mechanisms, they concluded that ORM proteins might have a secondary regulatory role on the sphingolipid biosynthesis pathway, which could be useful for engineering biotic stress tolerant crops. Finally, in the discussion, they suggest further experimental studies to improve the current models and to validate the model-generated regulatory mechanisms, hypothesized in this study.

The manuscript is well written and easy to follow, with only minor spelling errors, which are provided in this review. The study is sound and uses a rigorous stepwise methodology to make predictions on the regulatory processes of sphingolipids. The conclusions could significantly be strengthened if the secondary regulatory processes of ORM proteins were experimentally validated.

Recommendations:

• It would be useful to provide a (supplementary) figure with a growth curve, comparing experimentally determined and simulated growth rates.

• Please distribute code (e.g., Matlab scripts, Github link) and make all necessary data (e.g., sbml models) available in order to be able to reproduce the results.

• In S1 file, could include GPR associations of the 24 genes linked to the 78 reactions.

• Please enumerate the 23 postulated schemes used to predict the regulation of the sphingolipid pathway. S3 file does not provide sufficient information for each scheme.

• As the authors mentioned in the discussion, this study lacks experimental validation of the model-generated secondary regulatory mechanism of sphingolipids (“ceramide-ORM-ceramide synthase (class II) regulatory interaction”). If possible, it would be good to include some experimental validation.

• What are some limitations of using this approach?

• Please include references to previously published genome-scale metabolic models of Arabidopsis and highlight the advantage of using a pruned sphingolipid biosynthetic pathway in this study.

Minor edits:

Title:

• What is the evidence that this hypothetical regulatory mechanism is shared across different plants? I advise the authors to use a more specific term in the title.

Abstract:

• SPT: first mention of abbreviation without prior description

Introduction:

• 26carbon: space missing

• Misspelling in sphingolipid: "sphinolipid biosynthetic pathway"

• Long chain bases (LCB) abbreviated twice (lines 67 and 153)

Results:

• Text states that the model comprises 78 reactions, but S1 file enumerates 77

• Line 265: "(see Table 1 for details)”: should be Table 2

Discussion:

• "Non-stationary isotope experiments will be conducted in future studies to determine the accuracy of this assumption and the results will be used to improve the model. "It is not clear who will perform these studies, but it would be good to rephrase it.

• Lines 382 – 387: same as above

Methods:

• 12 to15-day-old : space missing

• (5 d, 10 d, 15d and 20 d): space missing between 15

• Line 441: it sounds like the sphingolipid components were measured in this study, although previously the authors mention that they were obtained from reference 16.

• Line 474: please capitalize Gibbs to be consistent with the remainder of text

• Line 483: "This procedure is repeated thousands of times ": please add the number of repetitions

• If available, please provide the humidity for the growth of the Arabidopsis seedlings.

Figures:

• Fig. 1 legend:

o What do the red flux values refer to?

o What is the unit of these fluxes, and where did they come from? The unit is mentioned in the S1 file.

• Fig. 2:

o The resolution could be improved

o The legend is insufficient to explain the figure

• Fig. 3:

o The resolution could be improved. The dashed reference line is not visible.

o Legend: define ORM

o “A” & “B” missing

Table 1:

• The reaction names are missing, it would be good to include them, at least in the caption.

Table 2:

• Define overexpression (OE)

• Reference to the asterisk of concentration missing

S1 file:

• What do the columns G & H refer to in the Excel spreadsheet?

• 77 reactions, but in the Results section 78 mentioned

S2 file:

• In the table, one of the ∆G'LB (kJ/mol) should be ∆G'UB (kJ/mol)

Reviewer #2: General comments

The manuscript reports efforts to apply the ensemble kinetic modeling approach to sphingolipid metabolism for testing hypotheses about the regulation of the sphingolipid pathway. A base kinetic model was parameterized using experimental data. With an independent set of criteria for model evaluation, individual models of regulatory hypotheses on top of the base model were evaluated and down selected. Repression of ceramides on orosomucoid proteins (ORMs) and activating interactions between ORMs and class II ceramide synthase (CSII) were identified to be the most promising hypotheses. The study constitutes a great example of integrating biochemical knowledge and experimental data for hypotheses testing and generation which can effectively suggest experimental targets, accelerating biological knowledge discovery.

Major comments

1. A set of reference steady-state metabolite and enzyme concentrations are needed for integrating the kinetic models (as also described in lines 487 – 488). What values did the authors select? Or are the steady-state fluxes and concentrations computed robust to the reference level?

2. A set of 23 regulatory schemes on top of the base kinetic model were compared. Were these models trained or fitted with parameters in any ways (any regulatory interactions in a kinetic model should need parameters)? If additional parameters were introduced for each model (hypothesis), how were the parameters fitted?

3. Following up, if additional parameters were introduced, do they truly better capture the evaluation criteria or simply because more introduced parameters lead to better fit (potentially overfitting)? Do the model pass for example Akaike information criterion or some likelihood-ratio test (since these are nested models based on the base model) with the log-likelihood modeled by for example cross entropy for the classification evaluation criteria used? This will give more confidence to the identified hypotheses.

4. Any additional interesting features by comparing the models that pass the filtering steps with the models that do not pass? This might generate new insights about how to engineer sphingolipid metabolism.

Minor comments

1. Table 2 shows the evaluation criteria. How well does each of the regulatory models perform? Will be good to provide a detailed table or the matlab scripts to reproduce this.

2. Also is the parameterized model made available?

3. Line 432: that should be sum(|v|) instead of simply sum(v)?

**Have all data underlying the figures and results presented in the manuscript been provided?**

Reviewer #1: **No: **Data for the Ensembl models and their perturbations are missing. It would be useful to include Matlab scripts and models in the System Biology Markup Language (SBML) format to be able to reproduce the results.

Reviewer #2: Yes

PLOS authors have the option to publish the peer review history of their article (what does this mean?). If published, this will include your full peer review and any attached files.

Reviewer #1: No

Reviewer #2: No
---

## [Decision Letter · Decision Letter 1]

14 Dec 2020

Dear Dr. Saha,

We are pleased to inform you that your manuscript 'Dissecting the regulatory roles of ORM proteins in the sphingolipid pathway of plants' has been provisionally accepted for publication in PLOS Computational Biology.

Best regards,

Kourosh Salehi-Ashtiani

Guest Editor

PLOS Computational Biology

Jason Papin

Editor-in-Chief

PLOS Computational Biology

Reviewer's Responses to Questions

**Comments to the Authors:**

Reviewer #1: I was pleased to read the revised version of the manuscript entitled “Dissecting the regulatory roles of ORM proteins in the sphingolipid pathway of plants”. The authors obviously invested a considerable amount of time and effort to address all major and minor recommendations. This study would be even more valuable if there was experimental validation of the predictions. I understand that the authors are in the process of performing such experiments and that more time is required. However, the current methodology used in this computational study is well defined and could be of relevance to the research community, not only for plant biology, but for other fields as well. I would like to thank the authors for making all the source code available on Github.

Reviewer #2: All the comments have been satisfactorily addressed. The reviewer in particular appreciates the additional efforts to analyze the kinetic parameters leading to some interesting insights.

**Have all data underlying the figures and results presented in the manuscript been provided?**

Reviewer #1: Yes

Reviewer #2: Yes

PLOS authors have the option to publish the peer review history of their article (what does this mean?). If published, this will include your full peer review and any attached files.

Reviewer #1: No

Reviewer #2: No

---

## [Editor Report · Acceptance letter]

22 Jan 2021

PCOMPBIOL-D-20-01501R1 

Dissecting the regulatory roles of ORM proteins in the sphingolipid pathway of plants

Dear Dr Saha,

I am pleased to inform you that your manuscript has been formally accepted for publication in PLOS Computational Biology. Your manuscript is now with our production department and you will be notified of the publication date in due course.

With kind regards,

Jutka Oroszlan
